# Competing social identities and intergroup discrimination: Evidence from a framed field experiment with high school students in Vietnam

Tam Kiet Vuong[1,2]☯*, Ho Fai Chan [1,2]☯*, Benno Torgler[1,2,3]☯

1 School of Economics and Finance, Queensland University of Technology, Brisbane, Australia, 2 Centre for Behavioural Economics, Society and Technology (BEST), Queensland University of Technology, Brisbane, Australia, 3 CREMA–Center for Research in Economics, Management and the Arts, Zurich, Switzerland

☯ These authors contributed equally to this work.
* vuongt@qut.edu.au (TKV); hofai.chan@qut.edu.au (HFC)

**Data Availability Statement:** Data and codes used in this study are available at OSF via https://osf.io/hd95a/ (doi: 10.17605/OSF.IO/HD95A).

## Abstract

We conducted a framed field experiment to explore a situation where individuals have potentially competing social identities to understand how group identification and socialisation affect in-group *favouritism and* out-group discrimination. The Dictator Game and the Trust Game were conducted in Vietnam's Ho Chi Minh City on two groups of high school students with different backgrounds, i.e., French bilingual and monolingual (Vietnamese) students. We find strong evidence for the presence of these two phenomena: our micro-analysis of within- and between-school effects show that bilingual students exhibit higher discriminatory behaviour toward non-bilinguals within the same school than toward other bilinguals from a different school, implying that group identity is a key factor in the explanation of intergroup cooperation and competition.

## Introduction

Identity is a source of pride and joy, confidence, and strength–but identity can also kill, as a strong and exclusive sense of belonging to a specific group can lead to a perception of distance to other groups or individuals [1]. For example, Amartya Sen [1] refers to the deadly Hindu-Muslim riots in the 1940s: "The political instigators who urged the killing (on behalf of what they respectively called "our people") managed to persuade many otherwise peaceable people of both communities to turn into dedicated thugs. They were made to think of themselves only as Hindus or only as Muslims (who must unleash vengeance on 'the other community') and as absolutely nothing else: not Indians, not sub-continentals, not Asians, not members of a shared human race" (p. 172). Both in-group favouritism and its opposite, out-group discrimination, are deeply rooted in human nature [2], for example, as Christakis [3] argues "the preference for one's own in-group is a cultural universal" (p. 266). Whereas in-group favouritism is a tendency to behave more favourably toward those of the same social group, who often

**Funding:** The author(s) received no specific funding for this work.

**Competing interests:** The authors have declared that no competing interests exist.

share similar values, interests, personal attributes, and characteristics [4–7], out-group discrimination is the withholding of such favouritism from those with whom the individual does not identify. In particular, in-group members may grant each other special privileges that they do not concede to outsiders [8, 9]. Two fundamentals crucial to in-group membership are familial and kinship networks, which serve to enhance sociability and sympathy while eliciting tolerance, fair play, and reciprocal obligations toward the well-being of others.

Identification is an important factor for living in a society [1]. Group members may feel they have a shared fate, a sense of togetherness and expectation of mutual aid or mutual-fate control [3]. A large number of experimental studies have shown ingroup biases since the pioneering work by Henri Tajfel and his co-authors in the 1970s (e.g., [10]) or the famous Robbers Cave camp experiment by Muzafer Sherif and his colleagues that powerfully demonstrated how intergroup hostility can emerge when groups are united against a common enemy [11]. Within such a context, previous experimental studies indicate that the willingness to adopt another person's attitudes and beliefs is strongly influenced by other members' in-group-out-group status (see, e.g., [12–17]). However, less well explored is that individuals also have *competing* identities. People constantly need to decide on the relative importance to attach to the respective identities [1].

To explore the impact of competing social identities, we will use a field experiment that allows us to observe individuals' choices when deciding on the relative importance to attach to various groups to which she or he belongs. Exploring their level of cooperation allows identification of the priority given by individuals to their various simultaneous identities. In general, the acceptance of a social identity can have traditional or cultural roots, but can also be shaped by daily activities, reorientation, and individual discovery. We therefore take advantage of a naturally unique setting in Vietnam, providing a cultural dualism that is particularly visible at the school level through two different teaching streams: French bilingual versus non-bilingual. Whereas all students share the same national Vietnamese curriculum, the French bilingual students are required to attend extra classes, participate in extracurricular activities, and sit extra examinations prior to Grade 1 and at the end of Grades 5, 9, and 12. Moreover, French bilinguals are part of a linguistic minority group. The French bilingual community in Vietnam is culturally, linguistically, and socially distinguishable compared with other groups of students and carries high entry and exit barriers to its programs. In addition, French bilinguals not only share the same curriculum but also socialise and participate in extracurricular activities together on a regular basis.

We therefore explore whether the bilingual students cooperate to the same or different degree with fellow bilingual members (from the same school and from a different school) and non-bilingual members within the same school. We predict that bilingual students exhibit stronger in-group favouritism and out-group discrimination due to the minority status of the group, which is consistent with the history of tribal welfare and ethnic conflict. This suggests that the tighter the community, the less value its members assigned to the humanity of strangers (p.47) [18]. However, strangeness is not an element in our setting, as we explore students' cooperative behaviour within students at the same school, with familiarity between students due to daily encounters and social interactions. However, to explore the linguistic or status signalling (level of prestige of the language track) aspect in more detail, we analyse not only how students within the same school (bilinguals and non-bilinguals) cooperate, but also how those students cooperate with bilingual students from another school.

Our study contributes to the literature on in-group biases as well as the literature on socialisation and social influence. Social influence has been extensively investigated in previous literature, exploring various dimensions of attitudes and beliefs; for example, stereotypes, prejudice, and political beliefs [16, 19–24]. However, less empirical attention has been given to

the question of whether and how socialisation has a significant impact on intergroup relations [25, 26]. Group socialisation means that individuals adopt norms, beliefs, values, and attitudes shared between members of that group [27–30]. In our setting, we study adolescents as they are in the process of learning how to become a member of society. Young people may find reasons to endorse or reject past traditions. They are required to make choices regarding their relevant identities and weigh the relative importance of these different identities, with potential for conflicting loyalties and priorities [1]. For example, in our case, decisions need to be made between the same commitments to the linguistic program or the same school. Previous literature suggests that such a learning process depends on the categorisation of self, which involves three main psychological stages of group formation: "(1) individuals define themselves as members of a distinct social category, (2) they form or learn the stereotypical norms of that category. . ., (3) they assign these norms to themselves. . .and thus their behaviour becomes more normative" [30] (pp. 72–73). From this perspective, social identification–that is, the extent to which individuals define themselves as members of a group–plays a central role in the psychological process of group socialisation [31].

Despite a wide-ranging body of research on intergroup discrimination and its economic implications conducted in diverse societies and vast arrays of dimensions (e.g., race, gender, religion, language), the number of studies on how cooperative and competitive behaviours develop in children and adolescents is still limited [32], even though the body of work examining cooperation in children and adolescents is growing (see, e.g., [33–36]). Hence, it is useful to increase knowledge in this area as effective policy implementation, allowing discriminatory behaviour to be addressed before it becomes internalised [37]. Current insights indicate that school-aged children seem to show more empathy for, expect more loyalty from, and grant more special favours to their in-group peers while expecting disloyalty from out-group opponents [38–40]. Given that trust toward strangers is an essential element in facilitating exchanges that boost economic development and prosperity [41–45], in-group biases can have substantial implications to society in terms of stability–due in part to its characteristic of durability–particularly in areas such as ethnicity or language. As [1] argues, "if, for example, she was to favour her own ethnic group in making public decisions, this could rightly be seen as a case of shady nepotism rather than an example of shining excellence of morality and ethics" (p. 32).

The remainder of the paper is structured as follows. Section 2 provides a brief historical background demonstrating the importance of the French language in Vietnam. Section 3 clarifies the study context by describing the characteristics of the two schools and sample groups. Section 4 then explains the experimental design. Section 5 reports the empirical results and Section 6 provides discussion and limitations. Finally, some concluding remarks are offered in Section 7.

## Historical background

### French in Vietnam

Vietnam and France's long-standing history began in the 17th century when Alexandre de Rhodes imported both the Roman alphabet and Christianity. From 1887 to 1940, under the French colonial *mission civilisatrice* ("civilising mission"), French was established as Vietnam's official language for education and social mobility. When Vietnam gained its independence from France on September 2, 1945, Vietnamese became the official national language, but French remained in use throughout French-controlled urban areas [46–48]. During the eight years of the resistance (1946–1954), new Vietnamese curricula were created and used in schools operating in demilitarised areas [49]. By 1954, Vietnam had become completely

independent from France, and the government began preparing for education reform aimed at reducing the number of years spent on general education [50].

Although only 0.5% of Vietnamese individuals still speak French today, the language's status lies in the two nations shared cultural and historical heritage [51]. For instance, many Vietnamese words are direct loanwords from French, while some famous Vietnamese dishes, such as *pho* or *banh mi*, combine the signatures of the two cuisines. Not only is Vietnam a member of the International Organization of the Francophonie (OIF), but France was also one of the first Western countries to support Vietnam's renewal policy (over a 20-year period). On September 25, 2013, the two nations signed a declaration of strategic partnership aimed at strengthening their relationship in all areas, including political, defence, economic, educational, and cultural [52].

### French in the education system

According to the French Embassy in Vietnam, the Vietnamese education system offers French teaching curricula on four distinct levels: major, intensive, minor, and bilingual classes (see Table A in S1 Appendix). This Vietnamese French bilingual education began in Vietnam in 1994 following the signing of international agreements between the governments of the former Indochina (Cambodia, Laos, and Vietnam), France, and the OIF [53]. These bilingual classes, taught in Vietnamese and French, require students to complete both 12-year curricula simultaneously [51]. Hence, to graduate from the bilingual program, students must sit two different examinations at the end of Grade 12: the Vietnamese National Baccalaureate and the French Bilingual Baccalaureate, which is recognised and accepted by the Francophone community. This type of bilingual education thus remains compatible with the traditional schooling system while providing cultural dualism [54, 55].

## School and subject characteristics

### Schools' characteristics

Ho Chi Minh City (commonly known as Saigon) is a centre of commerce located in southern Vietnam. The city, which played a significant role during the Vietnam War, is also known for its French colonial landmarks, including the Notre-Dame Cathedral and the 19th century Central Post Office (for city statistics, see Table B in S1 Appendix). Currently, however, only three public high schools in Ho Chi Minh City offer a bilingual program: *Marie Curie*, *Minh Khai*, and *Le Hong Phong*. Because *Le Hong Phong* is targeted specifically to gifted children, who are likely to have specific goals and unique characteristics, its inclusion in the study would lead to selection bias. We therefore focus only on the first two schools, *Marie Curie* and *Minh Khai*, both built by the French in the last century and among the oldest high schools in the city. Both institutions are in District 3, which places them only a 6-minute drive (1.1 km) away from each other and an 18-minute drive (3.8 km) away from District 1, the centre of Ho Chi Minh City. Because of this geographic proximity, the students at both schools tend to share similar socio-demographic and socio-economic backgrounds.

The focus of our study is the *Marie Curie* High School (known as *Lycée Marie Curie* in French and *Truong Trung Hoc Pho Thong Marie Curie* in Vietnamese), initially established by the French colonial government as an all-girls school and named after the female Polish-French Nobel Laureate [56–58]. At that time, all classes were conducted in French, with the majority of students being girls from French families and only a few local Vietnamese from wealthy or government employee families likely to enrol. The curriculum thus often specialised in subjects popular in Europe but undeveloped in Vietnam.

After finally admitting boys in 1970, the school was handed over to Ho Chi Minh City Education and Training Development in 1975, at which time the French teachers returned home. In 1997, the school was changed to a semi-public model and, with over 5,000 students per year, was once the largest high school in Vietnam. In the past 20 years, however, it has reduced its enrolment to increase the quality of its education. Our comparison school *Minh Khai*, founded in 1913, is also a public high school that began as an all-girls school (42 female students on its 1915 opening) with all classes conducted in French. After becoming coeducational in 1978, it has increased its enrolment numbers over the years through the inclusion of boarders and students from other provinces.

## Students characteristics

Each year, the French bilingual program offers a limited number of admissions to its future students. For instance, only 315 first grade bilingual students in Ho Chi Minh City were recruited in 2018, compared with over 90,000 grade 1 non-bilingual students [59]. At the end of grades 5, 9 and 12, bilingual students usually swap schools depending on their academic performance in the final examinations. Additionally, the French-language students study, socialise and participate in extracurricular activities together, which are mainly hosted by IDECAF–an organisation that specialises in promoting French culture, lifestyle, language, and cinematography to the Vietnamese community. Furthermore, as part of the Vietnam-France relations, bilingual teachers and students receive significant support from the French Embassy in Vietnam, including professional training, scholarships, annual festivals, and exchange programs [60].

According to the Embassy of France in Vietnam, 100% of French bilingual students obtain the Vietnamese National Baccalaureate, and 90% of them pass the National University Entrance Examination for admission, compared with 30% of single-language stream students who pass the admission exam. In addition, statistics collected from the University Agency of La Francophonie (AUF) surveys indicate that most Francophone students find employment in their first year of entering the labour market. For example, PFIEV–a high-quality engineering training first launched in France in the early 1990s –has been transferred to Vietnam to continue teaching and training high-qualified engineers. Students in this program can choose to learn French via an intensive mode to advance their studies in France. Around 60% of PFIEV graduates find work within three months. Furthermore, Vietnam-France cooperation in the health sector is also given priority and prominence among collaborative projects, including professional competence and the use of French language [61].

Table 1 presents characteristics of Marie Curie bilingual versus non-bilingual students from our post-experimental survey. We find statistically significant differences in gender, religion, and level of Westernisation between the two groups of participants. First, 81% of non-bilinguals are female, compared to 49% in the case of bilingual subjects. In addition, single-language children in this study are more religious than their bilingual counterparts (73% versus 55%). To estimate subjects' level of Westernization, we asked if they have been on exchange to France and whether they are planning to study overseas after graduation.

The results indicate statistically significant differences between the two groups, with 20% of bilingual students going on exchange to France, compared to only 3% for non-bilingual subjects. Furthermore, 54% of bilingual subjects reported planning to go overseas, whereas it was only 7% in the case of their counterparts. Even though we find no difference in average family income between the two studied groups, having plans to study overseas could be a significant indicator of a bilingual subject's social-economic background. For instance, only 0.1% of the

**Table 1. Characteristics of bilingual versus non-bilingual students.**

| | Bilingual (BC) participants | | Non-Bilingual (NC) participants | | | |
|---|---|---|---|---|---|---|
| | Mean | Std. Err. | Mean | Std. Err. | Diff. | z-stat. |
| Born in 2001 | 0.45 | 0.06 | 0.56 | 0.06 | -0.11 | -1.27 |
| Female (%) | 0.49 | 0.06 | 0.81 | 0.05 | -0.32*** | -3.99 |
| Atheism (%) | 0.45 | 0.06 | 0.27 | 0.05 | 0.18* | 2.18 |
| Marie Curie Pride (Scale 1–7) | 5.48 | 0.17 | 5.34 | 0.15 | 0.14 | 0.65 |
| Family Attachment (Scale 1–7) | 5.060 | 0.220 | 5.460 | 0.200 | -0.400 | -1.36 |
| Family income (%) | | | | | | |
| Above Average | 0.03 | 0.02 | 0.03 | 0.02 | 0 | 0.01 |
| Average | 0.83 | 0.05 | 0.81 | 0.05 | 0.01 | 0.18 |
| Below Average | 0.14 | 0.04 | 0.16 | 0.04 | -0.01 | -0.2 |
| | | | | | $\chi^2(2) = 0.04, p = 0.98$ | |
| Pocket money (%) | | | | | | |
| Below VND300,000 | 0.16 | 0.04 | 0.34 | 0.06 | -0.18* | -2.49 |
| VND300–500,000 | 0.49 | 0.06 | 0.29 | 0.05 | 0.21* | 2.5 |
| VND500,000 - 1MIL | 0.12 | 0.04 | 0.19 | 0.05 | -0.07 | -1.15 |
| Above 1MIL | 0.23 | 0.05 | 0.19 | 0.05 | 0.05 | 0.67 |
| | | | | | $\chi^2(3) = 9.95, p = 0.019$ | |
| Level of Westernisation (%Yes/No) | | | | | | |
| France exchange—Yes | 0.2 | 0.05 | 0.07 | 0.03 | 0.13* | 2.26 |
| Study overseas after graduate—Yes | 0.54 | 0.06 | 0.07 | 0.05 | -0.16* | -1.99 |

Test of proportion (z-test) was performed on all variables except for *Marie Curie Pride* and *Family Attachment*, for which the Mann-Whitney test was used. Sample size equals to $N_{BC}$ = 69 and $N_{NC}$ = 70.

[a] † $p < .10$

* $p < .05$

** $p < .01$

*** $p < .001$.

Vietnamese population could afford to study abroad, according to statistics reported from HSBC in 2016 [62].

## Experimental design

### Game selection

Our assessment tool is a set of Dictator and Trust Games that measure how group affiliation changes the participants' behaviours toward their in-group versus out-group members. In constructing our experimental design, we were inspired by the specialisations in two previous studies: [63] and [64]. Both studies employed the Trust game (along with other games) to measure discrimination and examine whether the trust allocations were due to taste-based or statistical discrimination (beliefs about trustworthiness). The concept of trust has been documented in various papers using participants from Asian regions (see [65–70]). In one of the relevant studies, the authors in [66] measured trust in institutions and generalised trust by using the data from the 2006 Asia Barometer survey and a two-item scale method in seven Confucian Asian countries, including Vietnam, China, Japan, Singapore, South Korea, Taiwan, and Hong Kong. The results showed that Vietnamese individuals exhibited the least trusting behaviours towards outsiders, where less than one out of five Vietnamese think people could be trusted.

## Subject pool

To facilitate identification/adherence of in-group versus out-group behavioral patterns, our subject pool consists of 70 *Marie Curie* bilingual students (BC), 72 *Marie Curie* non-bilingual students (NC), and 67 *Minh Khai* bilingual students (BK) in Grades 11 and 12. There are two classes per grade in each school. For *Marie Curie* students, we selected all four classes in which to distribute the invitations to participate: two bilingual and two non-bilingual groups in Grades 11 and 12 (ages from 16 to 17) who share the same timetable for extracurricular activities and sports. On the other hand, *Minh Khai* (bilingual) students only participated as *Receivers* (in the Dictator Game) or *Trustees* (in the Trust Game). As we had very limited access to the second school, we sent out invitations and conducted experiments in three of the bilingual classes (also in Grades 11 and 12). All invited students agreed to participate during break time. Also, it is worth noting that *Minh Khai* bilingual classes have smaller numbers of students (ranging from 21 to 23 per class, with two classes per grade) than *Marie Curie* bilingual classes. Thus, the subject pool covers most of the available grade 11 and 12 bilingual cohort at this school.

## Treatments

The first treatment, *PURE IN-GROUP*, includes only two in-group cases (BC-BC and NC-NC) at *Marie Curie*. Designed to measure the levels of in-group cooperation within two different streams at the same school, the two sessions under this treatment serves as a benchmark for the out-group treatments. The second treatment, *IN-GROUP STRANGER*, explores the first out-group condition by asking *Marie Curie* participants to interact with members of different language streams. This treatment also comprises two sessions with bilinguals non-bilinguals from the *Marie Curie* playing against each other (i.e., BC-NC and NC-BC). In the former session (BC-NC), bilingual participants serve as the first movers in the Trust and Dictator Games and the roles are reversed in the latter session (NC-BC), where non-bilingual participants play the roles of the *Trustor* and *Dictator*. This treatment thus encompasses in-group and out-group elements, with non-bilingual participants serving as an out-group to the *Marie Curie* bilingual group while still being students at the same school, a condition that should promote a certain level of closeness and familiarity. The third treatment, *OUT-GROUP*, conducted with only bilingual students from *Marie Curie* and *Minh Khai* (a single session of BC-BK), assesses whether in-group–out-group bias stems from the strong identification between members of the French bilingual groups. That is, although *Marie Curie* bilingual students might see *Minh Khai* bilingual students as an out-group from a different school, the two groups share certain in-group elements as members of the same bilingual community. In this treatment, therefore, the *Marie Curie* bilingual students acted as *Senders/Trustors* and the *Minh Khai* students as *Receivers/Trustees*.

## Game structure

Our experiment was approved by the QUT University Human Research Ethics Committee (QUT Ethics Approval Number 1700000762). Written consent was obtained from all participating students prior to the experiment. Additional parental consent was also obtained from the Marie Curie students due to the extended time commitment required for the experiment. The experiment proceeded as described below, using a game structure and experimental protocol very close to that employed in [71], where the authors investigated cultural integration versus convergence. Specifically, the protocol consisted of four commonly used economic games played using plain pen and paper: a Dictator Game, a Trust Game, a Risk Game, and a Competition Game. Notably, although our experiment included the Competition game used

in [71] and [72], only around 10% of the participants chose the competitive rate, so we do not explore those results in this study.

**Dictator game.** In the Dictator Game, participants were paired randomly, with each pair having a *Sender* (Player 1) and a *Receiver* (Player 2). The *Sender* received an initial endowment of VND 60,000 (10,000 Vietnamese Dong (VND) equals approximately $0.06 AUD. VND 60,000 = AUD $3.45) and had to decide how much of it (in $x$ whole numbers) to keep and how much to send to an anonymous receiver. The *Receiver* had no decision to make but simply accepted the amount awarded by the *Sender*, with final payoffs of 60,000 −$x$ and $x$ for the *Sender* and *Receiver*, respectively. Whereas BK students only participate as *Receiver*, each BC and NC participant played as both *Sender* and *Receiver* in the first two treatments [73–75], meaning that each made an allocation decision to one (anonymous) partner and received an allocation from a different (anonymous) partner. No two participants were paired twice during the entire experiment to preserve the one-period nature of the game. Specifically, BC students played it three times: with BC, NC, and BK students (as *Sender*). NC played it twice: with BC and other NC students, and BK once (as *Receiver* with BC students).

**Trust game.** Each pair in this game was made up of an *Allocator/Trustor* (Player A) whose behaviour indicated level of trust, and a *Recipient/Trustee* (Player B), whose behaviour indicated level of perceived trustworthiness. Each *Allocator* was endowed with VND 40,000 (equivalent to AUD$2.30) and had the option of keeping everything or sending a certain amount $x$ to an anonymous *Recipient* (where $0 \leq x \leq 40,000$). The amount that Player A sent to a recipient was tripled by the experimenter before being given to Player B. Player B then had the opportunity to keep all the money (sent from Player A) or send some or the entire amount $y$ back to Player A (where $0 \leq y \leq 3x$). The payoffs for Players A and B were therefore 40,000−$x$ + $y$ and 3$x$− $y$, respectively. As in the Dictator Game, each BC and NC player in the first two treatments made decisions as both *Allocator* and *Recipient*, whereas BK participants serve as *Recipient*s. No two participants were paired twice during the experiment.

**Risk game.** *The Risk Game was played by each Marie Curie participant to control for risk preferences when analysing the Trust experiment. It* was played once in each treatment following the Trust Game. Interestingly, decisions made the second time were identical to the first instance, which may indicate the validity of such a risk measurement due to its consistency. It may also indicate that subjects are able to recall information. Following [71] and [76], each participant was endowed with VND 20,000 (equivalent to AUD$1.15) and had to decide how much (from zero to all) to risk on an investment. The probability of tripling the amount invested was 50% but with a corresponding 50% chance of losing everything. The outcome was decided at the end of the experiment by flipping a coin.

## Experimental procedure

The four sessions at *Marie Curie* (BC-BC, NC-NC, BC-NC, and NC-BC) were conducted from 2:00 PM to 3:30 PM (Vietnam time) on September 12, 2017. The sessions were carried out in a large conference room where all *Marie Curie* participants could see one another. Each student was assigned a unique and random ID number for the experimental procedure, and provided with a set of general instructions prior to participation (see S2 Appendix). At the start of the first two treatments, all subjects were verbally informed that they would be play against another student unindentified to them in each Game, randomly drawn from the group appropriate to the treatment. That is, each *Marie Curie* student was paired with another student from the same (different) language stream in the first (second) treatment, respectively. We do this to make sure that students' decisions were not made based on the relationship towards a

specific individual. Furthermore, participants were told that they will be paried with a different student for each Game within each session.

The session pairs in the first two treatments were conducted simultaneously, e.g., BC-BC and NC-NC sessions happen at the same time. All participants completed the first treatment (*PURE IN-GROUP*) before moving on to the second treatment (*IN-GROUP STRANGER*). Notably, students were only informed about the group from which their 'opponent' was drawn from at the start of each treatment. As per experimental design, the third treatment (*OUT-GROUP*) only applied to bilingual students (BC-BK) to record their decisions (as the first mover) in the Dictator and Trust Game. Marie Curie bilingual students were told that they would be interacting anonymously with another fellow bilingual from *Minh Khai* High School. Similarly, participants from *Minh Khai* High School (all bilingual students) were also aware that they were paired anonymously with bilingual students from Marie Curie High School. The instructions were read aloud by the experimenter while the participants read along.

Before engaging in the experiment, the participants were informed that they would play four tasks (numbered 1 through 4 as part of the within-subject design). Marie Curie High School students were also informed that the four tasks would repeat for three (BC students) or two (NC students) times due to the experimental design. In addition to receiving general instructions for the experiment, before each game, the students were given envelopes containing a set of specific game instructions and forms for recording their decisions (see S2 Appendix). Issuing and collecting these forms in envelopes minimised self-presentation (for a discussion, see [71]). After the game instructions had been read aloud, the participants began completing the comprehension questions on their forms and recording their decisions. The instructions were in Vietnamese. No indication of language difficulties was identified, as all subjects share one native language, namely Vietnamese. Also, they were asked to complete the comprehension questions to ensure they understood the instructions before proceeding to the Game. Once all students had finished recording their answers, the envelopes were collected before those for the next game were distributed. The sequence of the four tasks presented to participants was the same for both treatments across all four sessions, which begins with the Dictator Game and is followed by the Trust Game, theRisk Game, and the Competition Game. Finally, a post-experiment survey was distributed that had to be completed before any game payouts could be made.

The overall payments consisted of a VND 50,000 show-up fee (equivalent to AUD$ 2.90), in addition to the amount earned during the experiment. All *Marie Curie* subjects received the same show-up fee as the required time commitment was the same (non-bilingual students have to wait in the same room for the bilingual students to complete their decisions in the third treatment). At the beginning of the experiment, *Marie Curie* participants were informed that the outcome of one of the games is chosen as the final payment, decided at the end of the experiment by a dice roll. If the Dictator Game or the Trust Game was selected, a coin was tossed to determine which role would be paid out (amount obtained as the *Sender* or *Receiver*). If the Risk Game was chosen, a coin toss determined the outcome: if heads, the participant received three times the amount invested; if tails, he or she lost the entire investment. Finally, if the Competition Game was selected for payment, the payoff depended on the participants' chosen option and actual performance. The show-up fee was made at the start of the experiment. The experimental incentives were paid the next day as the Risk Game was selected (at random) for the experimental payout. Participants receive the payment using the assigned ID numbers as verification.

The experimental session at *Minh Khai*, which forms the other half of the observations for the third treatment, took place in a spare classroom during school recess from 2:00 PM to 2:30 PM (Vietnam time) on September 19, 2017. In both the Dictator and Trust Games, these

participants served only as *Receivers* of the envelopes in which the *Marie Curie* bilingual students had recorded their *Sender* decisions. Hence, in the Dictator Game, the *Minh Khai* participants simply had to accept any amount sent by the *Marie Curie* bilinguals, but in the Trust Game, they had the opportunity to decide whether to keep the entire amount or send some back to an anonymous partner from the *Marie Curie* bilingual group. Prior to the experiment, the *Minh Khai* bilingual participants were informed that they would be paid VND 30,000 as a show-up fee in addition to any amount earned during the experiment. *Minh Khai* students received a lower show-up fee due their lower commitments in terms of time and efforts: 30 minutes compared to one and half hours for *Marie Curie* students. As with the *Marie Curie* session, the show-up fees were paid at the beginning of the experiment, while the incentive payments were made the following day based on ID numbers. In this treatment, the *Minh Khai* participants took home any amount received in the Dictator Game and any amount not sent back in the Trust Game. Notably, two *Marie Curie* non-bilingual students (selected at random) did not participate in the *IN-GROUP STRANGER* treatment due to the difference in number of students between language stream. Similarly, as the number of Marie Curie bilingual students was higher than the Minh Khai bilinguals (70 versus 67), we discarded all observations that did not have a paired partner in the third treatment prior to the analysis.

## Results

### Dictator game allocation

We find substantial evidence that in-group favouritism and out-group discrimination exist in both the bilingual and non-bilingual student groups. As indicated in Fig 1 and Table 2, we find that participants were more generous to their peers from the same language stream than to those from the other language stream within same school (pooled sample paired *t*-test: two-tailed $p < 0.001$), with the mean difference in amount offered of VND 18,037 (out of VND 60,000). Such difference is evident in both bilingual (mean difference = VND 21,071, $p < 0.001$) and non-bilingual (mean difference = VND 15,000, $p < 0.001$) participants. In addition, we also show the pairwise mean comparison between sessions and the results of Wilcoxon rank-sum tests (see Table A in S3 Appendix) with *p*-values adjusted for multiple comparisons (Bonferroni's method).

While the symmetry in behaviour is remarkable as both groups discriminate against each other substantially, bilingual students also showed stronger out-group discrimination than non-bilingual students. While the amount offered to peers within the same group (PURE IN-GROUP treatment) do not differ significantly between bilinguals and non-bilinguals ($p = 0.153$), bilingual participants offered significantly less to the non-bilingual counterparts ($p = 0.019$, IN-GROUP STRANGER treatment). Additionally, the overall amount of transfer to NC students was also less than that to BC students (mean difference VND 2,777, $p = 0.09$). The larger paired differences among bilinguals in the amount offered between the two groups ($p = 0.015$, PURE IN-GROUP minus IN-GROUP STRANGER) also shows that bilingual students are relatively less generous towards out-group, i.e., exhibiting more discriminatory behaviour compared to non-bilingual students. Interestingly, however, the bilingual *Marie Curie* students were also more generous to the bilingual *Minh Khai* students than to their non-bilingual peers from the same school (BC-NC vs BC-BK: mean difference of VND 12,239, $p < 0.001$; also, only 3% of the transfers to BK were zero), which may indicate that the language stream acts as a stronger in-group social identifier than the school.

In addition, as Fig 2 shows, whereas *Marie Curie* bilingual students transferred nothing to non-bilingual students in as many as 73% of cases (panel c), they sent zero to bilingual peers in less than 6% of cases (panel a). Only 19.9% (panel c) of non-bilinguals received an amount

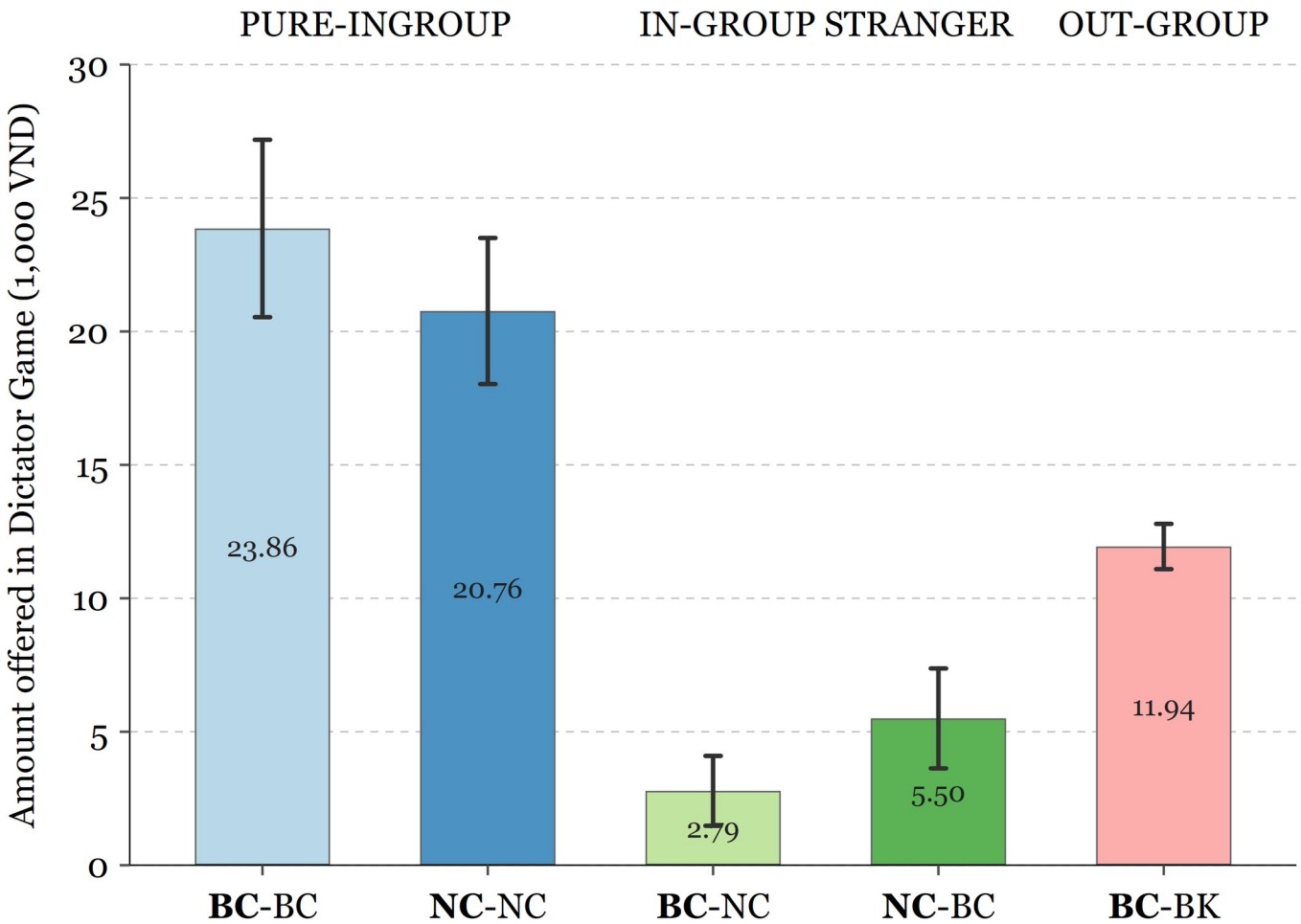

**Fig 1. Amount offered in Dictator Game by treatment and session.** Bar height indicates the average amount offered to Player 2. Error bars represent 95% confidence intervals. *Sender* (Player 1) in bold.

over VND 5,000 compared with 94.3% (panel a) and 97% (panel e) of *Marie Curie* and *Minh Khai* bilinguals, respectively. Conversely, whereas *Marie Curie* non-bilinguals sent more than or equal to VND 20,000 (one-third of the initial endowment) to about 66.8% (panel b) of their

**Table 2. Summary statistics of the amount offered in Dictator Game, by treatment and session.**

| Treatment | Mean | SD | Min | Max | N |
|---|---|---|---|---|---|
| *PURE IN-GROUP*: | | | | | |
| **BC**-BC | 23.86 | 13.94 | 0 | 60 | 70 |
| **NC**-NC | 20.76 | 11.65 | 0 | 60 | 72 |
| *IN-GROUP STRANGER*: | | | | | |
| **BC**-NC | 2.79 | 5.49 | 0 | 30 | 70 |
| **NC**-BC | 5.50 | 7.86 | 0 | 40 | 70 |
| *OUT-GROUP*: | | | | | |
| **BC**-BK | 11.94 | 3.48 | 0 | 20 | 67 |

Initial endowment of VND 60,000 (~AUD $3.45). *Sender* (Player 1) in bold.

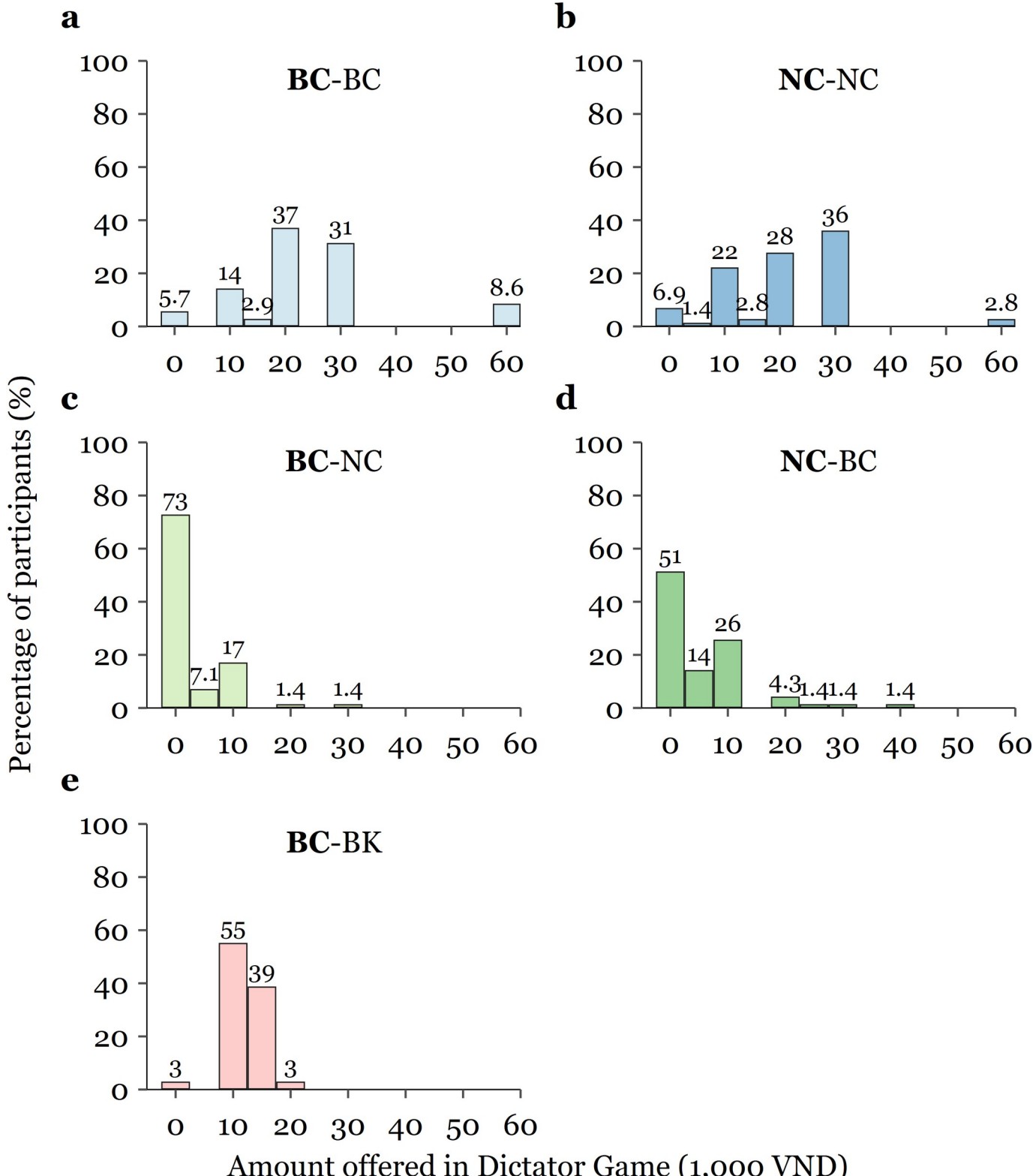

**Fig 2. Distribution of amount offered in Dictator Game, by treatment and session.** Bin width of VND 5,000. Panels a and b: PURE IN-GROUP treatment; Panels c and d: IN-GROUP STRANGER treatment; Panel e: OUT-GROUP: treatment. *Sender* (Player 1) in bold.

non-bilingual school peers, non-bilinguals allocated such a sum to only 8.5% (panel d) of their bilingual co-players.

## Trust game

The results for the Trust Game show similar favouritism and discrimination between groups (see Table 3 and Fig 3).

Overall, we find both bilingual and non-bilingual *Marie Curie* students transfer a larger sum to their peers within the same language stream than to the other stream (mean difference of VND 25,678, $p < 0.001$). Specifically, *Marie Curie* non-bilingual students show more trust in their non-bilingual peers ($M$ = VND 27,986) than in bilingual students at their own institution ($M$ = VND 2,643). Similarly, *Marie Curie* bilinguals exhibit higher trust in their school language peers ($M$ = VND 28,428) than non-bilingual students ($M$ = VND 2,357). Like the outcome of the Dictator Game, we also find that *Marie Curie* bilinguals placed more trust in *Minh Khai* bilinguals ($M$ = VND 6,418) than their non-bilingual peers. It should also be noted that all within-subject paired differences mentioned above are statistically significant at the 1% level. The difference in the amount transferred between treatments is also striking as about

**Table 3. Summary statistics of the amount sent and returned in Trust Game, by treatment and session.**

| Treatment | Mean | SD | Min | Max | N |
|---|---|---|---|---|---|
| Amount offered by **Player 1** | | | | | |
| *PURE IN-GROUP* | | | | | |
| **BC**-BC | 28.43 | 12.23 | 5 | 40 | 70 |
| **NC**-NC | 27.99 | 14.16 | 0 | 40 | 72 |
| *IN-GROUP STRANGER* | | | | | |
| **BC**-NC | 2.36 | 2.51 | 0 | 5 | 70 |
| **NC**-BC | 2.64 | 2.51 | 0 | 5 | 70 |
| *OUT-GROUP* | | | | | |
| **BC**-BK | 6.42 | 5.35 | 0 | 20 | 67 |
| Amount returned by **Player 2** | | | | | |
| *PURE IN-GROUP* | | | | | |
| **BC**-BC | 33.93 (33.93) | 24.66 (24.66) | 0 (0) | 70 (70) | 70 (70) |
| **NC**-NC | 38.96 (41.25) | 23.15 (21.72) | 0 (0) | 60 (60) | 72 (68) |
| *IN-GROUP STRANGER* | | | | | |
| **BC**-NC | 0.07 (0.14) | 0.6 (0.85) | 0 (0) | 5 (5) | 70 (35) |
| **NC**-BC | 0 (0) | 0 (0) | 0 (0) | 0 (0) | 70 (38) |
| *OUT-GROUP* | 5.67 (7.92) | 7.48 (7.78) | 0 (0) | 40 (40) | 67 (48) |
| BC-**BK*** | | | | | |
| Percentage returned by **Player 2** | | | | | |
| *PURE IN-GROUP* | | | | | |
| **BC**-BC | 38.89% | 20.01 | 0% | 77.78% | 70 |
| **NC**-NC | 43.64% | 13.66 | 0% | 66.67% | 68 |
| *IN-GROUP STRANGER* | | | | | |
| **BC**-NC | 0.95% | 5.6 | 0% | 33.33% | 35 |
| **NC**-BC | 0% | 0 | 0% | 0% | 0 |
| *OUT-GROUP* | 28.99% | 24.31 | 0% | 100% | 67 |
| BC-**BK*** | | | | | |

* Amount sent back by *Minh Khai* students (BK). Statistics on the amount returned given receiving non-zero amounts are in parentheses.

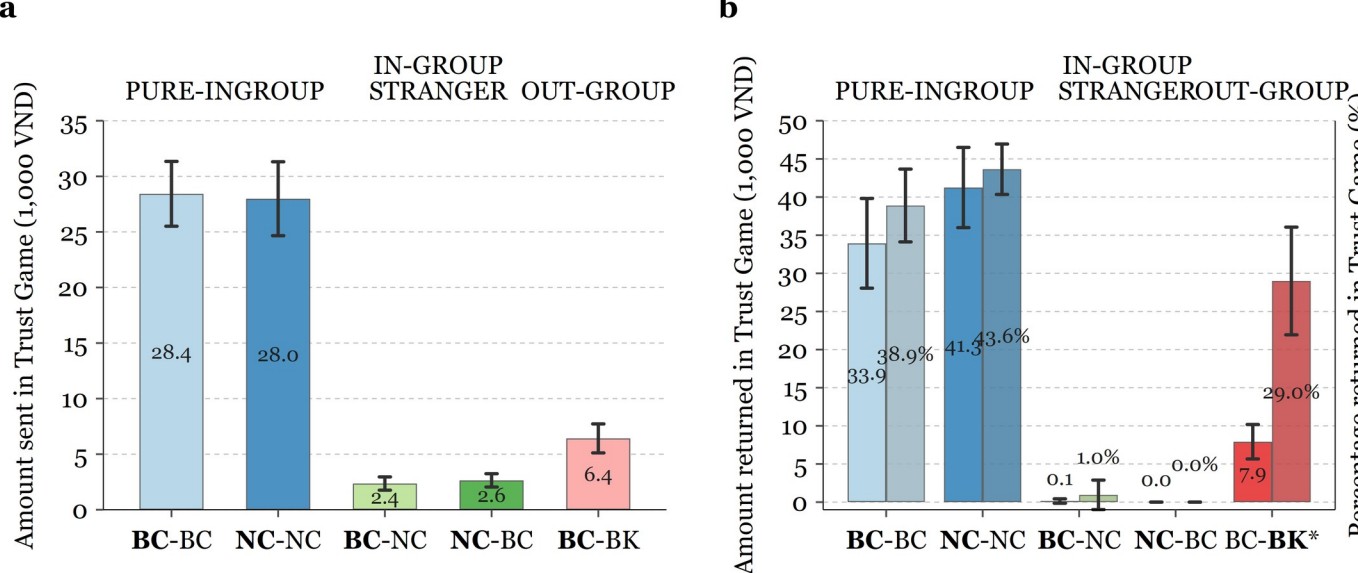

**Fig 3. Amount transferred in the Trust Game, by treatment and session.** Bar height indicates the average amount sent by Player 1 (panel **a**) and the average amount returned (lighter bars), and percentage returned (darker bars) by Player 2 given non-zero amount received (panel **b**). Error bars represent 95% confidence intervals. * Transfer by *Minh Khai* students (BK).

half of the students transferred the efficient amount (all VND 40,000) to their peers (Fig 4A and 4B), while no students made such choice when they were faced with students from different language stream (Fig 4C and 4D) or school (Fig 4E).

Nevertheless, the degree of (dis)trust to peers within the same language stream relative to the other language stream is similar across *Marie Curie* bilinguals and non-bilinguals. We did not find the amount transferred differ significantly between the two in both the PURE IN-GROUP (BC-BC versus NC-NC) and IN-GROUP STRANGERS treatments (BC-NC versus NC-BC) (see Table B in S3 Appendix for full pairwise comparisons between sessions (*t*-test and rank-sum test) with *p*-values adjusted for multiple comparisons). The *difference* in the amount transferred to Player 2 between in-group and out-group peers within the same school is also not statistically significant different between *Marie Curie* bilingual and non-bilingual students ($p = 0.731$). This result contrast with the finding of a larger degree of favouritism and discrimination among *Marie Curie* bilingual students in the Dictator Game.

With respect to *Trustees*, *Marie Curie* bilingual students sent more money back to their language program counterparts ($M = $ VND 33,929) than to their non-bilingual peers from their school (Table 3), even keeping the entire endowment for themselves when interacting with non-bilinguals (except for one who returned VND 5,000 among the 35 students who received a non-zero amount in the first stage). Similarly, whereas non-bilingual students returned no money to their bilingual peers at the same school, most of them transferred money back to Player 1 when faced with other non-bilingual students ($M = $ VND 41,250 for those who received non-zero amount). Interestingly, while the amount entrusted to same language stream peers within the same school do not differ across the two language streams, non-bilinguals seem to return a slightly larger share of the amount received (43.6% of the tripled sum) to their peers than do bilingual students (38.8% of the tripled sum), although the statistical significance of the difference is above the 10% threshold ($p = 0.107$). Comparatively, about 73% of the *Minh Khai* bilinguals (who received a non-zero sum) did send money back to their bilingual counterparts at the other school ($M = $ VND 7,917, about 29% of the tripled sum) (see

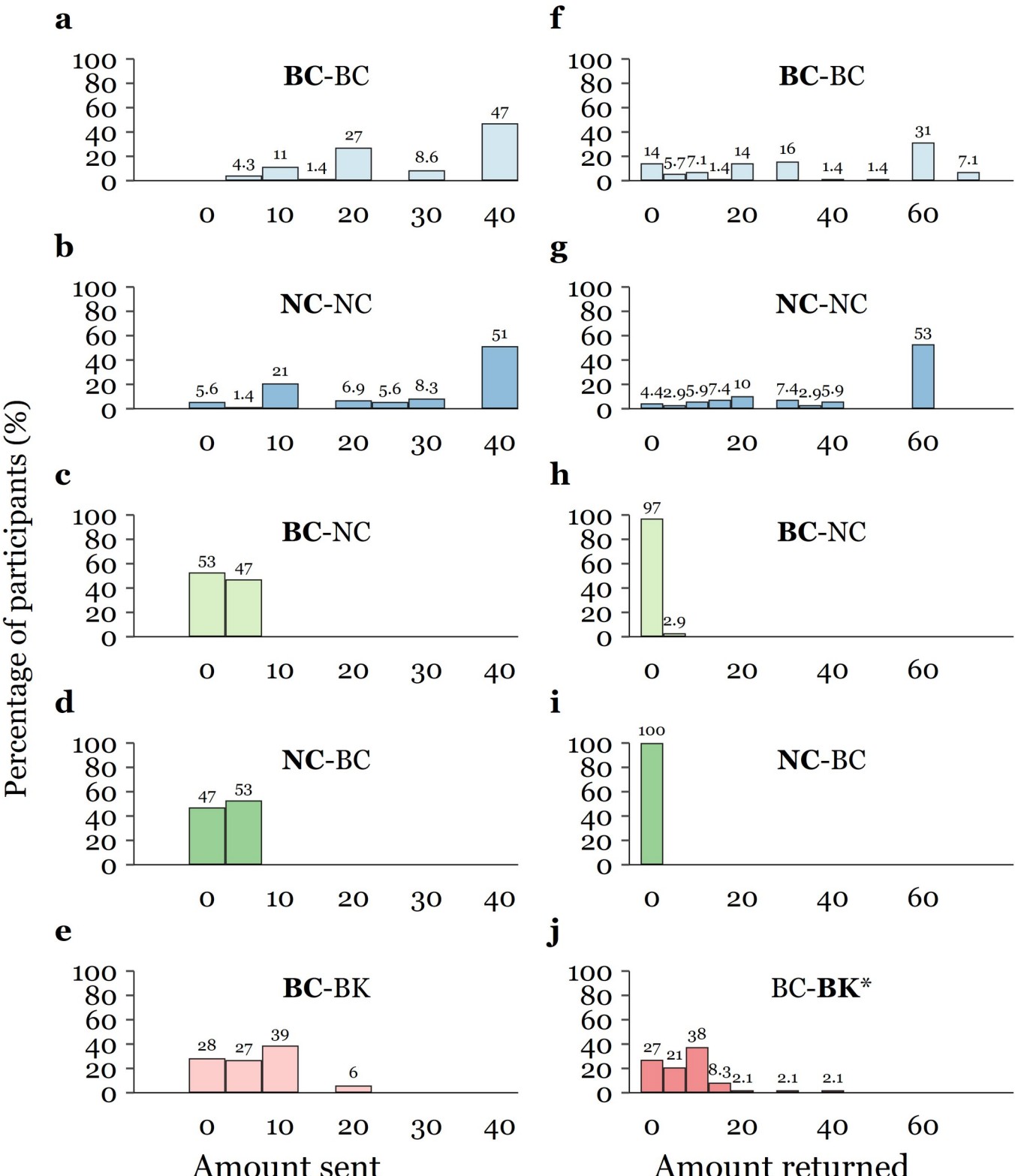

**Fig 4. Distribution of amount offered and returned in the Trust Game, by treatment and session.** Amount sent by Player 1 (panel **a** to **e**) and returned by Player 2 given non-zero amount received (panel **f** to **j**). * Amount sent back by *Minh Khai* students (BK).

**Table 4. Difference in money offered by Player 1 between the Dictator Game and Trust Game.**

| Treatment | Mean | SD | Min | Max | N | *t-stat.* | *p-value* |
|---|---|---|---|---|---|---|---|
| Percentage difference in amounts offered by **Player 1** between the Dictator Game and Trust Game | | | | | | | |
| *PURE IN-GROUP* | | | | | | | |
| **BC**-BC | -31.31 | 35.81 | -100 | 75 | 70 | -7.32 | < 0.001 |
| **NC**-NC | -35.36 | 36.01 | -100 | 50 | 72 | -8.33 | < 0.001 |
| *Between BC and NC* | 4.05 | | | | 142 | 0.67 | 0.503 |
| *IN-GROUP STRANGER* | | | | | | | |
| **BC**-NC | -1.25 | 11.16 | -12.5 | 50 | 70 | -0.94 | 0.352 |
| **NC**-BC | 2.56 | 14.07 | -12.5 | 54.17 | 70 | 1.52 | 0.133 |
| *Between BC and NC* | -3.81 | | | | 140 | -1.78 | 0.078 |
| *OUT-GROUP* | | | | | | | |
| **BC**-BK | 3.86 | 14.73 | -33.33 | 33.33 | 67 | 2.14 | 0.036 |

Table C in S3 Appendix for full pairwise comparisons in the amount returned between sessions (*t*-test and rank-sum test) with *p*-values adjusted for multiple comparisons.

## Decomposing other-regarding preferences and trust

In the Dictator Game, Player 2 cannot make any decision other than to accept the amount offered by Player 1, whereas in the Trust Game, Player 2 can decide how much to return (any part of the tripled amounts sent to them). Thus, Player 1's Trust Game choice could be motivated by trust or strategic decision making (i.e., hoping that Player 2 will return some amount (also depending on the amount sent) and altruistic reasons (other-regarding preferences), whereas trust does not play a role in the Dictator Game [64]. As we can observe the same subjects playing the Dictator as well as the Trust Game, we examine the percentage difference in the amounts transferred by Player 1 between the two games to elicit the motivating reasons behind choices in different sessions of the experiment. We present the results in Table 4. Notably, we use the percentage of the amount offered because of the different endowment amounts in the Dictator Game and Trust Game, VND 60,000 and VND 40,000, respectively. For example, if a player sent VND 20,000 (33.3% of VND 60,000) to the receiver in the Dictator Game and VND 20,000 (50% of VND 40,000) to the trustee in the Trust Game, the percentage difference in the amount transferred in the two Games is equal to -16.7 percentage points. The comparison is performed within subject and treatment (e.g., percentage amount transferred in Dictator and Trust Game from BC to NC).

The results suggest that the percentage amount offered by Player 1 is substantially higher in the Trust Game than in the Dictator game, for both bilingual (**BC**-BC) interactions (difference: 31.3 percentage points, $p < 0.001$) and non-bilingual (**NC**-NC) ones (difference: 35.4 percentage points, $p < 0.001$). This indicates that trust or strategic elements matter beyond unconditional other-regarding preferences, a pattern that does not differ between the two homogenous language streams ($p = 0.503$). In contrast, we do not find any statistically significant differences in the percentage of money offered when comparing the Dictator and Trust Game in mixed settings; this was the case for both the bilingual (**BC**-NC, $p = 0.352$) and the non-bilingual (**NC**-BC, $p = 0.133$) students. Considering that most of the offers in both Games were very low in this treatment, the insignificant results may be attributed to the lower bound problem in the form of distrust *and* absence of other-regarding attitudes towards the other group. Therefore, these findings confirm that in those interactions, students do not demonstrate enough trusting behaviour. There is a small difference of -3.81 percentage points between the two groups (statistically significant at 10% level, $p = 0.078$), which may suggest that non-bilingual students

report slightly higher distrust when trying to isolate other-regarding preferences. In addition, when Marie Curie bilinguals are faced with students from the Minh Khai school's bilingual language program (**BC**-BK), they transferred slightly more (3.86 percentage points) in the Dictator Game than in the Trust Game ($p = 0.036$) which shows that altruistic preferences are the dominant factor in that interaction.

## Regression analysis

In Table 5, we report the results of the Tobit model regressions (left-censored at 0) analysing the interactions among *Marie Curie* students in the Dictator Game (in terms of percentage of initial endowment transferred as Player 1) and Trust Game (percentage of initial endowment transferred as Player 1 and percentage of money returned as Player 2). Specifically, we examine the difference in decisions made between *Marie Curie* bilingual (BC) and non-bilingual (NC) students towards peers from the same and different language stream while controlling for socio-demographic and socio-economic factors. For the Trust Game decisions, we also control for the share of the money received in the Dictator Game as it was played prior to the Trust Game. Here, the *IN-GROUP STRANGER* variable indicates decisions made by participants *towards students from the other language stream* (**BC**-NC and **BC**-BC), with **BC**-BC and **NC**-NC (*PURE IN-GROUP*) as the reference group.

On average, and holding other factors constant, the *Marie Curie* participants gave 42.3% less to members of different language streams than to those in their own language stream at their own institution in the Dictator Game (specification (1)). Bilingual *Marie Curie* students also tended to give less on average than non-bilingual *Marie Curie* students, although the coefficient is not statistically significant. In specification (2), we include an interaction term to assess whether *Marie Curie* bilingual students (BC) are more likely than non-bilinguals to discriminate against other language streams, explicitly document that bilinguals sent 17.1% less to other language stream members than did non-bilinguals, indicating higher levels of out-group discrimination. However, we no longer observed a statistically significant interaction effect when the standard errors were clustered by classes instead of the individual level. This indicates that the decisions are likely to be correlated within some classes.

Further, we do not find a gender difference in the amount transferred in the Dictator Game, nor is the effect of participants age and religious belief statistically significant. However, the fact that NC has a larger female composition (as opposed to the more gender balanced bilingual class) would mean that in the IN-GROUP STRANGER treatment, BC students would expect to be paired with a NC female student 80% of the time while NC would expect to encounter either gender randomly when pairing with BC. This also means that NC would expect to be paired with another female NC more frequently than BC in the PURE IN-GROUP treatment. Thus, the interaction effect between BILINGUAL BC and IN-GROUP STRANGER might capture partially the effect in which women gives significantly less to other women while men's decisions are less dependent on the recipient. For example, while women are more generous than men in Dictator Game in general (e.g., [77]), some studies find that the gender difference is absent when the recipient is male or of unknown gender while female offer less to other females (e.g., [78–80] for an overview of the literature on gender difference in preferences).

Nevertheless, in an unreported regression including an additional interaction term with female (i.e., triple interaction terms between BILINGUAL BC, IN-GROUP STRANGER, and female), we do not find evidence that female and male BC students discriminate NC students statistically differently. In contrast, female NC made less transfer towards other NC students (relative to male NC and BC students) and slightly more towards students from the other

**Table 5. Within-school interaction (BC and NC).**

| | (1) | (2) | (3) | (4) | (5) | (6) |
|---|---|---|---|---|---|---|
| | Dictator (% Sent) | Dictator (% Sent) | Trust (% Sent) | Trust (% Sent) | Trust (% Returned) | Trust (% Returned) |
| **IN-GROUP STRANGER** | -42.3*** (3.59) | -34.0*** (4.00) | -69.5*** (5.24) | -69.6*** (6.53) | | |
| **Bilingual BC** | -2.48 (3.05) | 4.80 (4.16) | -6.32† (3.51) | -6.38 (5.09) | -4.14 (3.52) | -3.28 (3.47) |
| **Bilingual BC *** **IN-GROUP STRANGER** | | -17.1** (6.45) | | 0.13 (7.04) | | |
| Female | 2.41 (3.24) | 2.42 (3.27) | -11.4*** (3.36) | -11.4*** (3.35) | 2.24 (3.64) | 1.04 (3.77) |
| Age | -1.07 (2.40) | -1.35 (2.42) | 0.17 (3.25) | 0.17 (3.25) | 6.68* (2.97) | 7.86** (2.86) |
| Atheism | 2.63 (3.13) | 2.65 (3.15) | 3.61 (3.42) | 3.60 (3.44) | -3.18 (3.33) | -2.90 (3.11) |
| *Family income:* | | | | | | |
| Below average | 22.8*** (6.79) | 22.7*** (6.80) | 0.24 (10.8) | 0.23 (10.8) | 2.55 (4.93) | 2.58 (4.09) |
| Above average | -0.61 (4.76) | -0.60 (4.74) | -3.75 (4.49) | -3.75 (4.50) | 5.24 (4.37) | 3.23 (4.43) |
| *Pocket money:* | | | | | | |
| Below 300k | -5.00 | -4.74 | -14.6** | -14.6** | -0.35 | -0.43 |
| | (3.47) | (3.44) | (4.40) | (4.40) | (4.03) | (3.79) |
| 500k-1 million | 3.91 | 4.21 | -11.6* | -11.6* | -1.45 | 1.64 |
| | (4.21) | (4.21) | (4.69) | (4.69) | (3.07) | (3.07) |
| Above 1 mil | 2.66 | 2.81 | -11.1* | -11.1* | -9.49† | -9.56* |
| | (3.87) | (3.90) | (4.62) | (4.61) | (5.08) | (4.82) |
| Risk (% invested) | | | -0.13* | -0.13* | | |
| | | | (0.058) | (0.058) | | |
| Dictator (% sent) | | | 0.30** | 0.30** | | |
| | | | (0.11) | (0.11) | | |
| Dictator (% received) | | | -0.071 | -0.071 | | -0.059 |
| | | | (0.11) | (0.11) | | (0.11) |
| Trust (% received) | | | | | | 0.18** |
| | | | | | | (0.057) |
| Constant | 52.5 | 53.3 | 86.5 | 86.5 | -66.9 | -97.2* |
| | (40.8) | (40.8) | (53.7) | (53.7) | (49.5) | (48.9) |
| Observations | 276 | 276 | 276 | 276 | 135 | 135 |
| N (cluster) | 139 | 139 | 139 | 139 | | |
| Pseudo R² | 0.089 | 0.093 | 0.132 | 0.132 | 0.014 | 0.025 |
| Prob. > F | 0.000 | 0.000 | 0.000 | 0.000 | 0.116 | 0.000 |

Columns (1)–(6) report the coefficients in the Tobit estimations. Standard errors are in parentheses; Standard errors are clustered over subjects for (1)−(4) and robust for (5)−(6). For specifications (5) and (6), trust return equals 0 for both IN-GROUP STRANGER treatments (BC−NC and NC−BC), so the regression excludes these observations. The reference groups are "Average" for family income and "300k-500k" for pocket money. Female and Atheism are dummy variables.

[a] † $p < .10$

* $p < .05$

** $p < .01$

*** $p < .001$.

language stream. Furthermore, we find that students with less than average family income are more generous in the Dictator Game than the reference group (Average) while pocket money has no significant effect.

To isolate the effect of *Trustor*'s trust beliefs (i.e., whether the amount entrusted would be returned) from their risk attitudes in the decision to send money to a *Trustee* in the Trust Game, in specifications (3) and (4), we include a control for risk behaviour (the % of the

amount invested in the Risk Game). Additionally, we also include their decision made as Player 1 in the Dictator Game to control for participant's altruistic preference towards the Trustee from the same group, as well as the amount participants received as Player 2 in the Dictator Game. The results are very similar: trustees in the BC-NC and NC-BC treatments received around 68% less money than in the in-group settings (BC-BC, NC-NC). BC participants again sent less on average than NC students (around 6.8%, statistically significant only at 10% level). However, they did not seem to discriminate more (place less trust) against individuals from the other language stream than do NC participants (i.e., the interaction effect is not statistically significant). Expectedly, altruistic attitudes are positively related to decisions to transfer more as Trustor. For instance, for every ten-percentage point additional transferred in the Dictator Game, Trustor sent 2.7 percentage points more in the Trust Game.

Surprisingly, risk attitude shows a negative effect on the amount sent in the Trust Game, which is in contrast to the findings of previous studies where no relationship was found between risk aversion and decisions in investment games (e.g., [81–83]) or positive relationship (e.g., [84]). We do not know why this might be the case. The share of the amount received in the Dictator Game does not seem to influence participants' trusting behaviour. Moreover, we find that female participants sent less money to the Trustee in general, which aligns with the prior research findings where males place more trust than females (e.g., [85]). Nonetheless, as suggested by the literature [85], the gender of the responder is not likely to affect the level of trust is placed between men and women in the Trust Game. Therefore, by controlling for gender, the main interaction term (between BC and the treatment variable) should not be cofounded by the gender effect. Similarly, we did not find age nor atheism being a significant factor in determining trusting behaviour.

Lastly, specifications (5) and (6) examine the participants' return behaviour in the Trust Game (see variable Trust (% Returned), which takes a value from 0 to 100). Because the amount returns equals zero for all participants receiving a non-zero amount from Player 1 in the IN-GROUP STRANGER treatment (i.e., NC-BC and only 1 exception for BC-NC), we exclude all observations from the IN-GROUP STRANGER treatment and focus on the difference in the amount returned to peers from same language stream between BCs and NCs. The result shows that BC participants gave less to in-group members, albeit the coefficients are not statistically significant, even controlling for the amount received by the *Trustor* (specification (6)).

In line with the extensive literature on the importance of reciprocity (see, e.g., [86]), receiving more from the *Trustor* is associated with trustees returning more money to the trustors (on average around 18% of the money received). While the proportions returned by females is higher, the effect is not statistically significant. The difference in gender composition among senders between BC and NC is not likely to affect the results as the sender's gender is not a significant factor in determining reciprocal behaviour [85]. Interestingly, we find that older participants return significantly more to the sender. Although all participants are in Grades 11 and 12 (ages from 16 to 17), the coefficients indicate that students in Grade 12 (age 17) return around 8 percentage points more than their younger counterparts. Lastly, we did not find a statistically significant effect on trustworthiness with regards to religious belief, amount received in the dictator game, and both socio-economic factors.

Next, in Table 6, we examine the results for between-school interactions, focusing particularly on the behaviour of bilingual students (with BC–NC session as the reference group). All factors held constant, in the Dictator Game, BC participants sent 47.5 percentage points more of the initial endowment to BC students than to NC students and 27.6 percentage points more to bilingual students from *Minh Khai* (BK). These observations, which suggest stronger in-group identification with language stream than with school attended, are confirmed by the

**Table 6. Between-school effects.**

| | (1) | (2) | (3) |
|---|---|---|---|
| | Dictator %Sent | Trust % Sent | Trust % Sent |
| **BC–BC** | 47.5*** | 75.6*** | 68.4*** |
| | (5.18) | (4.36) | (6.75) |
| **BC–BK** | 27.6*** | 15.9*** | 12.8** |
| | (3.67) | (3.78) | (4.33) |
| **Female** | 5.16† | -5.58 | -6.35† |
| | (2.63) | (3.47) | (3.57) |
| **Age** | -6.92** | -1.54 | -0.48 |
| | (2.65) | (3.48) | (3.58) |
| **Atheism** | 1.39 | 3.21 | 3.74 |
| | (2.70) | (3.37) | (3.28) |
| **Family income:** | | | |
| **Below average** | 11.5 | 5.32 | 7.87 |
| | (11.3) | (7.22) | (5.96) |
| **Above average** | -0.13 | -5.57 | -5.88 |
| | (4.02) | (6.65) | (6.34) |
| **Pocket money:** | | | |
| **Below 300k** | -0.23 | -2.88 | -5.18 |
| | (4.24) | (4.87) | (4.54) |
| **500k-1 million** | 7.57* | 0.14 | -1.69 |
| | (3.18) | (4.73) | (4.99) |
| **Above 1mil** | -0.88 | -5.86 | -7.36† |
| | (2.90) | (4.16) | (4.28) |
| **Risk (% invested)** | | | -0.13* |
| | | | (0.061) |
| **Dictator (% sent)** | | | 0.21 |
| | | | (0.16) |
| **Constant** | 102.2* | 24.3 | 17.3 |
| | (43.3) | (57.9) | (59.8) |
| Observations | 204 | 204 | 204 |
| N (cluster) | 69 | 69 | 69 |
| Pseudo $R^2$ | 0.108 | 0.128 | 0.132 |
| Prob. $> F$ | 0.000 | 0.000 | 0.000 |

Columns (1)–(5) report the coefficients in the Tobit estimations. Standard errors, given in parentheses, are clustered over subjects. The reference groups are BC–NC for (1) to (3) and BC–BK in (4) and (5), "Average" for family income and "300k-500k" for pocket money. Female and Atheism are dummy variables.

† $p < .10$

* $p < .05$

** $p < .01$

*** $p < .001$.

results for the Trust Game in which bilingual *Marie Curie* participants (BC) sent 15.9% more on average to *Minh Khai* bilinguals (BK) than to *Marie Curie* non-bilinguals (NC). Even though BC students have no information on the gender composition of the group of BK students (one might expect the ratio to be similar as BCs), the difference between the Dictator Game (altruistic) transfers to NC and BK could also be attributed to female-to-female discrimination [77]. Nevertheless, this should not explain the difference observed in the Trust Game.

## Discussion

The findings of this paper provide evidence to support the role of group socialisation in inter-group cooperation and competition. On the basis of the self-categorisation theory which suggests that self-defining social categorisation is a fundamental precondition involved in the psychological process of group affiliation [87], such perceptions toward insiders as "us" and outsiders as "them" lead to discrimination in favour of the in-groups [10, 88, 89]. That is, when subjects define themselves as members of the group, they tend to internalise the shared attitudes and beliefs during their socialisation process and, involve 'private acceptance of a norm which defines a group in which subjects include themselves and with which they identify' [12] (p.207). As expected, the French bilingual members succeeded in maintaining a strong and separate group identity, where they displayed greater in-group favouritism to fellow bilinguals from a different school than the non-bilingual schoolmates. The findings of this paper are also aligned with previous studies, which indicate that individuals are more positively attracted to outsiders that share the same beliefs, values, or personality characteristics than outsiders that are dissimilar to their own [90–93].

Several reasons led us to believe that Marie Curie French bilingual subjects cooperate differently with respect to fellow bilingual students than with non-bilingual members. First, the significant magnitude of the observed discrimination may account, in part, for the longevity of the French history in Vietnam and its apparent success at maintaining common-pool resources. Generally stated, the French bilingual members selectively adjusted their cooperativeness according to the in-group characteristics of the person with whom they interacted. In the second (*IN-GROUP STRANGER*) and third (*OUT-GROUP*) treatment, the Marie Curie bilingual subjects displayed the tendency to be less favourable toward the non-bilingual schoolmates than to the Minh Khai bilingual members. From the shared experience and frequent interactions, the bilingual subjects might have learned that cooperation tends to be reciprocated. Hence, even though they all remained anonymous throughout the experiments, the willingness to cooperate with fellow bilinguals with whom they share a common fate–a fate that is dependent on cooperation, was greater than the interaction with non-bilinguals.

Second, the role of group socialisation has a significant effect on intergroup attitudes and beliefs. The French bilingual students are often perceived as holding more prestige than other students, particularly with respect to higher socio-economic backgrounds and academic performance. This may contribute to the relatively stronger discriminatory choices observed among French bilingual students in the Dictator Game than their peers in the non-bilingual program. Furthermore, as the chosen age groups are from 16 to 17 years old (grade 11 and 12), these bilingual participants consider their fellow members as a primary social group and are expected to exhibit the strongest group conformity due to their longest time associated with the group (from school-age to adolescent development stages) [94].

A third explanation for the discriminatory behaviour is that we might have observed the bilingual subjects playing a super-game. As the two studied schools are very close to each other, the participants might be concerned about how their decisions would impact future interactions with fellow bilingual members, thus adjusting their cooperativeness accordingly. This concern did not exist when paired with a non-bilingual member.

Finally, another interesting result was drawn outside of the bilinguals' behavioural tendencies. In the second treatment *IN-GROUP STRANGER*, the symmetry in behaviours was remarkable, presenting evidence that both studied groups discriminated against each other equally although to a slightly varying degree in the Dictator Game. In other words, this was not only the case of high-status students discriminating against others but could also be explained by the differences in perspectives of minority versus majority groups (e.g., see [86]).

Therefore, we see several potential avenues for future research. First, extending the investigation by adding another out-group condition (e.g., NC–NK) and across different age groups (as in [32]) would help to examine another layer of intergroup discrimination dynamic and thus, help to understand the driving forces behind discriminatory behaviour. Second, it would be interesting to explore a bilingual-centric variation to examine whether the in-group favouritism shared among bilingual subjects is consistent when paired with fellow bilingual members from a different city (e.g., Northern versus Southern Vietnam). In addition, information from the survey can be used for further research, such as exploring how in-group-out-group trust questions, child qualities or Big 5 personality traits could affect favouritism and discrimination.

The findings of this study must be seen in light of some limitations. First, it should be mentioned that not using a strategy method when eliciting trustworthiness of Receivers comes at a cost. It reduces the possibility of assessing whether the level of trustworthiness differs depending on whether subjects respond to bilingual or non-bilingual senders, and thus whether some of the differences in trust may reflect accurate expectations of differences in trustworthiness. The issue is that, given that Senders transfer very different amounts to bilingual and non-bilingual receivers, receivers then start with very different endowments and thus could be comparing amounts sent back. Second, *Marie Curie* participants were not explicitly told ex ante that they would interact with students from another language stream (or from a different school) at a later stage before playing the *PURE IN-GROUP*. Thus, participants' decisions in the first treatment may have been framed less towards an in-group/out-group scenario. Likewise, participants may potentially exhibit less out-group discrimination if the treatments were ordered differently (e.g., reversed) or if a randomisation in the treatment orders would have been implemented. Third, as the experiment was conducted during the normal class timetable, we were under some pressure to complete the experiment on time due to the busy schedule of year 12 students and the bilingual cohort. Hence, the experimental procedure used at Marie Curie High School was a mix between verbal and written instructions while making sure that the verbal instructions were fully understood. Finally, one should be careful in generalising the results as the data only came from one single school, as the importance of a particular identity depends on the social context. For example, different schools may be characterised by different levels of teachers reinforcing the higher status of bilingual educational tracks. Therefore, more diverse school profiles and bigger sample size can be beneficial for future studies.

## Concluding remarks

The core innovation in this study is to identify a setting in which competing identities are expected, and then to identify the level of in-group favouritism and out-group discrimination among school-aged children, along with a broader set of social preferences including altruism, trust, and trustworthiness. We applied a framed field experiment exploiting the unique language and cultural background of Ho Chi Minh City, Vietnam, to conduct an analysis of within-school and between-school effects among French bilingual versus non-bilingual high school students. With respect to the intergroup discrimination dynamics, we present two important findings. First, the symmetry in behaviour is remarkable as both groups discriminate against each other equally, and thus very low levels of sharing and trusting were recorded. Second, the bilingual students exhibit higher levels of discriminatory behaviour toward non-bilinguals within the same school (*Marie Curie*) than other bilinguals from a different school (*Minh Khai*). Such consistent differences (regardless of their reasons) may indicate a potential for disharmony. History is full of examples of campaigns attempting to emphasise singular identities; triggering such in-group biases can lead to atrocities, making even old friends into new enemies.

In general, cultural diversity can bring benefits to society by providing ample variety of experiences to enjoy, but cultural diversity does not mean support for cultural conservatism that may enhance ingroup biases or reduce cultural freedom [1]. As Amartya Sen in [1] points out, the "merit of diversity must thus depend on precisely *how* that diversity is brought about and sustained" (p. 116). He emphasises that even if certain basic cultural attitudes and beliefs influence the nature of reasoning, they cannot invariably determine it fully (p. 34). Thus, giving people an opportunity for exercising freedom is important. The violation of freedom can result in a lack of knowledge and understanding of other cultures and alternative lifestyles. The fact that in-group biases are seen at the school provides further support for the need to discuss human identities at the school level; according to [1], this is where schools can play a critical role: "There is . . . the important recognition that human identities can take many distinct forms and that people have to use reasoning to decide on how to see themselves, and what significance they should attach to having been born a member of a particular community. . . a person may well decide that her ethnic or cultural identity is less important to her than, say, her political convictions, or her professional commitments, or her literary persuasions. It is a choice for her to make, no matter what her place is in the strangely imagined 'federation of cultures'" (pp. 119, 159).

## Supporting information

**S1 Appendix. School characteristics.**
(PDF)

**S2 Appendix. Experimental instructions.**
(PDF)

**S3 Appendix. Additional results.**
(PDF)

## Acknowledgments

The authors would like to thank Manh Hung Nguyen, Vice Principal and Ngoc An Thi Nguyen, Head Teacher of Foreign Language Department at Marie Curie High School and Trieu Doan, Head Teacher of Foreign Language Department at Nguyen Thi Minh Khai High School for their approval and assistance in conducting the field experiments of this study. Moreover, for helpful comments and suggestions thanks are due to two anonymous referees and the PLOS ONE academic editor Natalia Jiménez.

## Author Contributions

**Conceptualization:** Tam Kiet Vuong, Benno Torgler.

**Data curation:** Tam Kiet Vuong.

**Formal analysis:** Tam Kiet Vuong, Ho Fai Chan, Benno Torgler.

**Funding acquisition:** Benno Torgler.

**Investigation:** Tam Kiet Vuong, Ho Fai Chan, Benno Torgler.

**Methodology:** Tam Kiet Vuong, Ho Fai Chan, Benno Torgler.

**Software:** Ho Fai Chan.

**Supervision:** Benno Torgler.

**Validation:** Ho Fai Chan.

**Visualization:** Ho Fai Chan.

**Writing – original draft:** Tam Kiet Vuong, Ho Fai Chan, Benno Torgler.

**Writing – review & editing:** Tam Kiet Vuong, Ho Fai Chan, Benno Torgler.

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
