## [Decision Letter · Decision Letter 0]

6 Oct 2021

PONE-D-21-25423Competing Social Identities and Intergroup Discrimination: Evidence from a Framed Field Experiment with High School Students in VietnamPLOS ONE

Dear Dr. Chan,

Thank you for submitting your manuscript to PLOS ONE. After careful consideration, we feel that it has merit but does not fully meet PLOS ONE’s publication criteria as it currently stands. Therefore, we invite you to submit a revised version of the manuscript that addresses the points raised during the review process.

I have selected major revision because the two reviewers have opposite recommendations. As the research question is appealing and the data analysis have been correctly performed, I think it's worthy to give you a chance as long as you address all the concerns raised by reviewer 1 about the experimental design which I share.

We look forward to receiving your revised manuscript.

Kind regards,

Natalia Jiménez, Ph.D. Economics

Academic Editor

PLOS ONE

Journal Requirements:

2. Please confirm the IRB approved the specific consent procedure used for students in this study.

4. We note that Figure A1 in your submission contain map images which may be copyrighted. All PLOS content is published under the Creative Commons Attribution License (CC BY 4.0), which means that the manuscript, images, and Supporting Information files will be freely available online, and any third party is permitted to access, download, copy, distribute, and use these materials in any way, even commercially, with proper attribution. For these reasons, we cannot publish previously copyrighted maps or satellite images created using proprietary data, such as Google software (Google Maps, Street View, and Earth). For more information, see our copyright guidelines: http://journals.plos.org/plosone/s/licenses-and-copyright.

 a. You may seek permission from the original copyright holder of Figure A1 to publish the content specifically under the CC BY 4.0 license. 

Additional Editor Comments:

Dear Authors,

I have selected major revision because one of the reviewers is very satisfied with your study while the other is not. Both seem to find the reserach question and the data analysis appropriate since they do not raise any concern about them. However, I agree with reviewer 1 that you should provide more details about the design and that you must justify why you give different information in the instructions to the 2 schools you consider. In your next submission, you have to address all the points raised by reviewer 1.

Reviewers' comments:

Reviewer's Responses to Questions

**Comments to the Author**

1. Is the manuscript technically sound, and do the data support the conclusions?

Reviewer #1: No

Reviewer #2: Yes

2. Has the statistical analysis been performed appropriately and rigorously? 

Reviewer #1: Yes

Reviewer #2: Yes

3. Have the authors made all data underlying the findings in their manuscript fully available?

Reviewer #1: Yes

Reviewer #2: Yes

4. Is the manuscript presented in an intelligible fashion and written in standard English?

Reviewer #1: Yes

Reviewer #2: Yes

5. Review Comments to the Author

Reviewer #1: The methods and procedures of the experiment have not been described in sufficient detail. Besides, the experiment might not have been conducted according to the technical standards.

It is nowhere specified how many sessions were conducted and how many participants were in each session. Were students of each class mentioned in section 4.2 conducting an independent session? Maybe a table could help to clarify this issue. Also, in page 12 lines 13-15 it is said “For Marie Curie High School students, they were also informed that the four tasks would repeat for three (BC students) or two (NC students) times, due to the experimental design.” but it is not specified the order of such repetitions and whether the order was randomized.

The main feature of the experiment that differentiates the treatments in this experiment is the matching protocol between participants. Looking at the experimental instructions provided in Appendix B, I can see that the experimental instructions provided to Minh Khai High School students for tasks 1 and 2 specify the matching protocol. However, this is not the case for Marie Curie High School students, whose instructions state, “Each of you will play this task with someone from this group.” I do not understand why at the Marie Curie sessions “all subjects were verbally informed that they would be paired with an anonymous member from the same and different language stream within Marie Curie High School, respectively.” (pag 12 section 4.5 line 4-6). In my opinion, each set of experimental instructions should have specified the particular matching a participant is involved in. This issue might be a serious problem in terms of accomplishing the technical standards of conducting experiments.

Reviewer #2: I enjoyed a lot reading the paper! I think that the authors do a fantastic job at motivating and writing it.

They conducted a framed field experiment to explore a situation where individuals have potentially competing social identities to understand how group identification and socialization affect ingroup favoritism and out-group discrimination. Despite all the evidence on this subject, this papers study teenagers which is relevant for understanding on how cooperative and competitive behaviors develop in children and adolescents, as it suggested by atuhors.

I believe that this paper could make a great publication for this journal.

6. PLOS authors have the option to publish the peer review history of their article (what does this mean?). If published, this will include your full peer review and any attached files.

Reviewer #1: No

Reviewer #2: **Yes: **Diego Jorrat

---

## [Author Response · Author response to Decision Letter 0]

26 Oct 2021

Editor Comments:

I have selected a major revision because one of the reviewers is very satisfied with your study while the other is not. Both seem to find the research question and the data analysis appropriate since they do not raise any concern about them. However, I agree with reviewer 1 that you should provide more details about the design and that you must justify why you give different information in the instructions to the 2 schools you consider. In your next submission, you have to address all the points raised by reviewer 1.

Response: We thank the Editor and both Reviewers for their valuable comments. We have now addressed all concerns raised by Reviewer 1 concerning the experimental design. Please see our responses below.

Reviewer #1: 

The methods and procedures of the experiment have not been described in sufficient detail. Besides, the experiment might not have been conducted according to the technical standards. It is nowhere specified how many sessions were conducted and how many participants were in each session (1). Were students of each class mentioned in section 4.2 conducting an independent session? (2) Maybe a table could help to clarify this issue. Also, in page 12 lines 13-15 it is said “For Marie Curie High School students, they were also informed that the four tasks would repeat for three (BC students) or two (NC students) times, due to the experimental design.” but it is not specified the order of such repetitions and whether the order was randomized (3).

Response: We have now provided more details on the experiment and have clarified all aspects raised by the Reviewer. Please see the following comments that deal with the specific points:

(1) The experiment consists of five sessions in total across the three treatments: PURE IN-GROUP (two sessions), IN-GROUP STRANGER (two sessions) and OUT-GROUP (one session). The number of participants in each session was: session 1 (BC- BC): 70 students, session 2 (NC-NC): 72 students, session 3 (BC-NC): 70 students, session 4 (NC-BC): 70 students and session 5 (BC-BK): 67 students (also reported in Table 2). Furthermore, we have also clarified that observations with no matched partner were discarded due to difference in the number of students across groups. 

(2) We have clarified this point in the Experimental Procedure section. For example, we added the text: “The four sessions at Marie Curie (BC-BC, NC-NC, BC-NC, and NC-BC) were conducted from 2:00 PM to 3:30 PM (Vietnam time) on September 12, 2017. The sessions were carried out in a large conference room where all Marie Curie participants could see one another” (see page 15). The experimental session with Minh Khai students was conducted a week later. We have therefore added the following text: “The experimental session at Minh Khai, which forms the other half of the observations for the third treatment, took place in a spare classroom during school recess from 2:00 PM to 2:30 PM (Vietnam time) on September 19, 2017” (see page 17).

(3) It is now stated that “The sequence of the four tasks presented to participants was the same for both treatments across all four sessions, which begins with the Dictator Game and is followed by the Trust Game and Risk Game, and finally, the Competition Game.” (see page 16). For the third treatment, Marie Curie bilingual students recorded their decisions for the Dictator Game first and then the Trust Game before their answers were transferred to Minh Khai bilingual students (one week later). The Minh Khai bilingual student treatment (Minh Khai bilingual students only play as the second mover in the Dictator and Trust Game) also follows the same sequence (Dictator=>Trust Game). In addition, we have now clarified the sequence of the treatments by providing now also a discussion on the implication of treatment orders (see page 36). 

The main feature of the experiment that differentiates the treatments in this experiment is the matching protocol between participants. Looking at the experimental instructions provided in Appendix B, I can see that the experimental instructions provided to Minh Khai High School students for tasks 1 and 2 specify the matching protocol. However, this is not the case for Marie Curie High School students, whose instructions state, “Each of you will play this task with someone from this group.” I do not understand why at the Marie Curie sessions “all subjects were verbally informed that they would be paired with an anonymous member from the same and different language stream within Marie Curie High School, respectively.” (page 12 section 4.5 line 4-6). In my opinion, each set of experimental instructions should have specified the particular matching a participant is involved in. This issue might be a serious problem in terms of accomplishing the technical standards of conducting experiments.

Response: 

We have now clarified the matching protocol for each treatment. The randomization process of pairing students is essentially the same across ALL treatments as each student was informed that they are paired with another unidentified students, drawn from the group specified by the treatment. That is, in the first treatment (PURE IN-GROUP), Marie Curie students will be paired with another MC student randomly drawn from the same language stream (and redrew for different Game). In the second treatment, MC students were told that they will now be matched with another (again, unidentified) student from another language stream. The wording “Each of you will play this task with someone from” was accompanied by the verbal explanation of the group in which the student was drawn from (subject to the treatment), i.e., ‘this group’ refers to ‘students from the same language stream’ in the first treatment, and ‘students from the different language stream’ in the second treatment. While communicating it verbally we made sure that all students fully understood the pairing. 

 As our subjects are high school students and the experiment was conducted during the school timetable, parents of Marie Curie students were informed to come and pick up their children at a specific time. Therefore, we were under some time pressure to complete the experiment within the given duration. To accommodate for this time constraint and to prevent any potential errors in distributing the tasks, we simplified the procedure by printing out the same instructions for the first and second treatments while carefully explaining to the students the matching procedure in each session. We now discuss this aspect in the limitation section. 

Reviewer #2: 

I enjoyed a lot reading the paper! I think that the authors do a fantastic job at motivating and writing it.They conducted a framed field experiment to explore a situation where individuals have potentially competing social identities to understand how group identification and socialization affect ingroup favoritism and out-group discrimination. Despite all the evidence on this subject, this papers study teenagers which is relevant for understanding on how cooperative and competitive behaviors develop in children and adolescents, as it suggested by atuhors. I believe that this paper could make a great publication for this journal.

Response: 

We are very thankful for the positive feedback! 

Journal Requirements

1. Comment: Please ensure that your manuscript meets PLOS ONE's style requirements, including those for file naming. 

Response: Thank you. We have updated the manuscript to ensure that the manuscript meets the journal style requirements. 

2. Comment: Please confirm the IRB approved the specific consent procedure used for students in this study.

Response: Yes, as part of the IRB approval, both schools provided written permissions for the experiment to be conducted using their student population. We have added a sentence regarding written consent in the Experimental Design section. In Section 4.4 we also provide the information that the experiment was approved by the QUT University Human Research Ethics Committee (including the approval number).

For Marie Curie students, as the time commitment required from them was higher (1.5 hours) and the experiment also took place during class timetable (which was normally allocated for sports and extracurricular activities), we asked for both, written parental and student consent prior to the experiment. 

On the other hand, for Minh Khai students, the time commitment was only 30 minutes, and the experiment took place during school recess which created almost no disruption or inconvenience for students. Therefore, only student consent was asked before they participated in the games. 

3. Comment: Please include captions for your Supporting Information files at the end of your manuscript, and update any in-text citations to match accordingly.

Response: We have completed the updates. Thank you. 

4. Comment: We note that Figure A1 in your submission contains map images which may be copyrighted. All PLOS content is published under the Creative Commons Attribution License (CC BY 4.0), which means that the manuscript, images, and Supporting Information files will be freely available online, and any third party is permitted to access, download, copy, distribute, and use these materials in any way, even commercially, with proper attribution. For these reasons, we cannot publish previously copyrighted maps or satellite images created using proprietary data, such as Google software (Google Maps, Street View, and Earth). For more information, see our copyright guidelines: http://journals.plos.org/plosone/s/licenses-and-copyright.

Response: Thank you for your suggestion. To avoid copyrights infringement, we have removed Figure A1 from the manuscript. As the figure was there for illustrative purposes only, removing it does not have any impact on the paper.

---

## [Decision Letter · Decision Letter 1]

26 Nov 2021

Competing Social Identities and Intergroup Discrimination: Evidence from a Framed Field Experiment with High School Students in Vietnam

PONE-D-21-25423R1

Dear Dr. Chan,

We’re pleased to inform you that your manuscript has been judged scientifically suitable for publication and will be formally accepted for publication once it meets all outstanding technical requirements.

Kind regards,

Natalia Jiménez, Ph.D. Economics

Academic Editor

PLOS ONE

Additional Editor Comments (optional):

Reviewers' comments:

Reviewer's Responses to Questions

**Comments to the Author**

1. If the authors have adequately addressed your comments raised in a previous round of review and you feel that this manuscript is now acceptable for publication, you may indicate that here to bypass the “Comments to the Author” section, enter your conflict of interest statement in the “Confidential to Editor” section, and submit your "Accept" recommendation.

Reviewer #1: All comments have been addressed

Reviewer #2: All comments have been addressed

2. Is the manuscript technically sound, and do the data support the conclusions?

Reviewer #1: Yes

Reviewer #2: Yes

3. Has the statistical analysis been performed appropriately and rigorously? 

Reviewer #1: Yes

Reviewer #2: Yes

4. Have the authors made all data underlying the findings in their manuscript fully available?

Reviewer #1: Yes

Reviewer #2: Yes

5. Is the manuscript presented in an intelligible fashion and written in standard English?

Reviewer #1: Yes

Reviewer #2: Yes

6. Review Comments to the Author

Reviewer #1: (No Response)

Reviewer #2: All my comments were addressed. Also recommendations of the other referee and the clarifications made by the authors, improve substantially the paper.

7. PLOS authors have the option to publish the peer review history of their article (what does this mean?). If published, this will include your full peer review and any attached files.

Reviewer #1: No

Reviewer #2: No

---

## [Editor Report · Acceptance letter]

2 Dec 2021

PONE-D-21-25423R1 

Competing Social Identities and Intergroup Discrimination: Evidence from a Framed Field Experiment with High School Students in Vietnam 

Dear Dr. Chan:

I'm pleased to inform you that your manuscript has been deemed suitable for publication in PLOS ONE. Congratulations! Your manuscript is now with our production department. 

Kind regards, 

on behalf of

Dr. Natalia Jiménez 

Academic Editor

PLOS ONE